# High Basal Maximal Standardized Uptake Value (SUV_max_) in Follicular Lymphoma Identifies Patients with a Low Risk of Long-Term Relapse

**DOI:** 10.3390/cancers13122876

**Published:** 2021-06-09

**Authors:** Giovanni Manfredi Assanto, Giulia Ciotti, Mattia Brescini, Maria Lucia De Luca, Giorgia Annechini, Gianna Maria D’Elia, Roberta Agrippino, Ilaria Del Giudice, Maurizio Martelli, Agostino Chiaravalloti, Alessandro Pulsoni

**Affiliations:** 1Hematology, Department of Translational and Precision Medicine, Sapienza University of Rome, Via Benevento 6, 00161 Rome, Italy; assanto@bce.uniroma1.it (G.M.A.); ciotti@bce.uniroma1.it (G.C.); brescini@bce.uniroma1.it (M.B.); deluca@bce.uniroma1.it (M.L.D.L.); annechini@bce.uniroma1.it (G.A.); delia@bce.uniroma1.it (G.M.D.); Agrippino@bce.uniroma1.it (R.A.); delgiudice@bce.uniroma1.it (I.D.G.); martelli@bce.uniroma1.it (M.M.); 2Nuclear Medicine, Department of Biomedicine and Prevention, University Tor Vergata, 00133 Rome, Italy; agostino.chiaravalloti@gmail.com; 3Nuclear Medicine, Istituto Neurologico Mediterraneo IRCCS Neuromed, 86077 Pozzilli, Italy

**Keywords:** follicular lymphoma, PET-SCAN, lymphoproliferative disease, lymphoma, SUV

## Abstract

**Simple Summary:**

A simple and easily available parameter, such as SUV_max_, could represent a useful tool in clinical practice to evaluate at diagnosis the risk of late relapse in Follicular Lymphoma. A higher basal FDG uptake (>6) was associated with a lower long-term relapse probability only in the absence of other risk factors (bone marrow involvement, B-symptoms, extra-nodal disease, elevated LDH, and/or b2-microglobulin), which can overwhelm the favourable effects of a high SUV. A low basal SUV_max_ reflects an indolent behaviour with a higher rate of late relapse, thus, requiring a prolonged follow-up.

**Abstract:**

Background: Despite that the unfavorable prognostic role of a high Total Metabolic Tumor Volume (TMTV) in Follicular Lymphoma has been demonstrated, the role of SUV_max_ alone at baseline PET/CT could have a different prognostic role. Patients and Methods: We performed a retrospective observational monocentric cohort study. All patients affected by FL who underwent a basal PET/CT were included. Two subgroups were identified and compared in terms of PFS and OS: (A) Basal SUV_max_ ≤ 6; and (B) Basal SUV_max_ > 6. Results: Ninety-four patients were included, 34 in group A (36.2%) and 60 in group B (63.8%). The PFS at two years was comparable in the two groups (97%). The five-year PFS was 73.5% for group A and 95% for group B (*p* 0.005). The five-year PFS in the whole cohort was 87.5%. A clear advantage was confirmed in group A in the absence of other risk factors. Patients with SUV_max_ ≤ 6 and no risk factors showed a 5-year PFS of 73% against 83% for patients with SUV_max_ > 6 and at least two risk factors. Conclusion: A high FDG uptake favorably correlated with PFS. A low basal SUV_max_ reflected a higher rate of late relapse requiring a prolonged follow-up. The basal SUV_max_ is an approachable parameter with prognostic implications.

## 1. Introduction

Follicular lymphoma is the most common indolent Non-Hodgkin Lymphoma (NHL) and the second most common subtype in Western countries. Although the combination of anti-CD20 antibody and chemotherapy remarkably improved the prognosis of FL patients, approximately 20% of patients relapse within two years from front-line therapy and have a poor prognosis, while for most patients, late relapse occurs. These patients are not easily identified at diagnosis by current prognostic scores, such as the Follicular Lymphoma International Prognostic Index (FLIPI) or FLIPI2 [1].

Two-deoxy-2-(18F) fluoro-D-glucose (18F-FDG) Positron Emission Tomography/Computed Tomography (PET/CT) is currently a standard imaging technology for diagnosis, staging, and response evaluation in patients with HL or NHL. It has been shown that, despite its indolent biology, follicular lymphoma is avid for 18F-FDG, and over 90% of patients are PET/CT positive at the initial presentation [2]. In FL, PET/CT can identify the disease even with a small tumor size and in a localized presentation [3,4].

Findings from prospective studies on high-tumor-burden FL treated with immunochemotherapy on the first line showed the prognostic role of 18F-FDG PET/CT performed at the end of treatment in terms of progression-free survival (PFS) and overall survival (OS). Post-Induction PET is, therefore, a reliable prognostic tool for identifying patients with a high risk of relapse [5,6,7]. Despite its role in staging and its predictive value in response evaluation, PET-based imaging’s prognostic role at the time of diagnosis needs to be better defined [8,9,10]. To identify poor-risk patients before initiating therapy, functional parameters, mainly the total metabolic tumor volume (TMTV) quantified on baseline PET, have recently gained interest.

TMTV refers to the tumor’s metabolically active volume, obtained by summing the metabolic volumes of all nodal and extra-nodal lesions. The European Association of Nuclear Medicine recommends the use of the 41% SUV_max_ threshold method. In a pooled analysis conducted on high tumor burden FL treated at diagnosis, TMTV showed a correlation with PFS and OS. Patients with TMTV <510 cm [3] had a five-year PFS and a five-year OS that were significantly greater than patients with a high TMTV [1]. Through multivariate analysis, the TMTV and FLIPI2 scores were independent predictors of PFS. In combination, they identified three risk groups: high TMTV and intermediate to- high FLIPI2 score with a 5-year PFS of 20, high TMTV, or intermediate-to-high FLIPI2 score with 5-year PFS of 46%; and low TMTV and low FLIP2 with 5-year PFS of 69% [11].

TMTV, thus, represents one of the most potent functional parameters among those currently available. However, the absence of a standardized method makes it not easily reproducible and consequently not routinely employed in clinical practice. Conversely, SUV_max_ is an easily available parameter largely used to assess disease activity that could produce further information of prognostic relevance, especially in low tumor burden disease [12].

In a retrospective analysis on 346 patients with advanced-stage FL, SUV_max_ > 18 was associated with significantly shorter PFS among patients treated with non-anthracycline-based regimens but not among patients treated with R-CHOP. SUV_max_ > 18 was also associated with shorter overall survival (OS) in patients treated with R-CHOP and non-anthracycline-based frontline regimens [13]. In another study conducted on 54 patients, the baseline SUV_max_ showed a significant association between B symptoms, the number of different lymph node sites involved, LDH, FLIPI, and TMTV. Furthermore, the univariate analysis demonstrated a correlation between SUV_max_ and PFS but not with OS [14].

Hypothesizing that a high metabolic activity could identify patients carrying a high risk disease, we investigated the prognostic role of SUV_max_ at basal PET/CT, considered as the SUV value at the site with the highest uptake of fluorodeoxyglucose (FDG) in terms of the PFS, OS, and event-free survival (EFS), considered from the start of treatment or diagnosis.

## 2. Materials and Methods

A retrospective observational monocentric cohort study was performed at the Hematology Department of the Sapienza University of Rome. All patients affected by FL who underwent a basal staging PET/CT between 2008 and 2018 were included. The internal review board approved this study. The study respects the ethical principles of the 2008 Helsinki Declaration.

### 2.1. PET/CT Scanning

The PET/CT system VCT (GE Medical Systems, Memphis, TN, USA) was used to assess the 18F-FDG distribution in all patients by 3D-mode standard technique in a 128 × 128 matrix. Reconstruction was performed using the 3-dimensional reconstruction method of ordered-subsets expectation maximization (OSEM) with 21 subsets and with two iterations. All the subjects in our study were injected with 2.5 MBq/kg ± 10% (210–410 MBq) of 18F-FDG i.v. and hydrated with 500 mL of i.v. saline sodium chloride (NaCl) 0.9%.

18F-FDG was injected in a dedicated room for each patient with lights off. All the patients were required to remain in resting conditions with their eyes closed prior to the PET/CT scan. A whole-body PET/CT scan was performed ~60 min after the 18F-FDG injection. A low-amperage whole-body CT scan for attenuation correction (40 mA; 120 Kv) was performed before PET image acquisition [15].

Values for the mean and max standard uptake value (SUV, g/mL) were calculated by an experienced nuclear medicine physician on a dedicated workstation (ADVANTAGE WORKSTATION 4.4. GE MEDICAL SYSTEMS) for all PET/CT examinations. A region of interest (ROI) was drawn on the pathological area that showed a higher uptake of 18F-FDG. After their positioning, all the VOIs were further checked, in a three planar view, by two experienced physicians to exclude unwanted tissues in the area of interest. The same methodology was previously used in a similar report from our group in this field [16].

### 2.2. Patient Selection

The diagnosis was based on histological examination with immunohistochemistry of lymph node and bone marrow (BM) biopsy). According to the WHO classification [17], 3B forms or concurrent diffuse large B-cell lymphoma (DLBCL) were excluded. All patients included in our analysis underwent diagnostic work-up according to current guidelines [18,19].

Patients were stratified according to the maximum SUV value assessed at the onset. The analysis was initially conducted with different cut-offs, according to the existing literature and specific receiver operating characteristic (ROC) curve using the basal SUV_max_ and events of relapse [12,13]. Finally, a cut-off of 6 points of SUV was identified as potentially significant with the best ratio between sensitivity and specificity (60% and 73% respectively) and the strongest association with progression free survival (OR 0.234; 95% IC 0.58–0.934; *p* 0.04). Therefore, two major significant subgroups were identified and related to data collected by review of medical records. The patients’ clinical and pathological features were registered and stratified as reported in Table 1. FLIPI and FLIPI2 scores were calculated for each patient [20,21].

Patients in whom immediate therapy was necessary were identified according to the GELF (Groupe d’Etude des Lymphomes Folliculaires) criteria [22]. Patients not fulfilling the criteria were not treated immediately, adopting a watch and wait strategy. Patients who fulfilled the GELF criteria were treated according to the guidelines and clinical judgment [18]. For localized FL, radiotherapy alone was performed. Advanced stage patients were treated mainly with R-CHOP/R-CHOP-like regimens (rituximab, cyclophosphamide, doxorubicin, vincristine, and prednisone), until 2015; afterward, a Rituximab–Bendamustine regimen was also largely employed [23]. Rituximab maintenance was administered in advanced stage FL, on clinical judgement. The treatment response was defined according to Lugano criteria [24]. The two subgroups: (A) Basal SUV_max_ ≤ 6; (B) SUV_max_ > 6; were compared in terms of the PFS and OS, and the correlation to each parameter is shown in Table 1.

### 2.3. Statistical Analysis

Statistical analysis was performed using IBM Statistics 25.0™. Continuous variables were summarized as the median and interquartile distance or the mean and standard deviation (SD). The categorical variables were expressed as absolute and percentage frequencies. Analysis of single groups was made following the D’Agostino–Pearson normality test. Categorical covariates were compared using Fisher’s exact test or the Chi^2^ test, if appropriate. The risk of the event was assessed as survival functions (PFS, OS, and EFS using the Kaplan–Meier method with the estimated 95% confidence interval (95% CI, standard error from Greenwood’s formula) in univariate and multivariate analysis. Multivariate analysis was conducted using the Cox Regression Model. Comparative tests for survival distribution were made with the Log-rank (Mantel–Cox), Breslow, and Taron–Ware tests, to assess the statistical significance.

## 3. Results

Ninety-four patients were included. The median age at diagnosis was 57 years (range 25–80), and 44.7% were male (44) and 55.3% (52) were female. The median follow-up was 60 months (range 15–139 months). At basal PET/CT, the median value of SUV_max_ was 8.2 (range 1.5–22.4). Thirty-four patients were included in group A (36.2%) and 60 patients were included in group B (63.8%).

In Table 1, we summarized baseline characteristics for each group. According to the univariate analysis of PFS in the two groups, Group A showed an inferior estimated median PFS: 92 vs. 122 months for group B (*p* 0.005) (Figure 1). The PFS at two years was not significantly different in the two groups, (Group A 96.8% vs. Group B 97.3%). Conversely, a significant difference in 5-year PFS was observed, resulting in 73.5% for group A and 95% for patients in group B (*p* 0.005) (Figure 1). The five-year PFS in the whole cohort was 87.5%.

In the whole cohort, the OS was 94.7%, and five deaths occurred, three of which were observed in Group A and two in Group B. No statistical difference was observed between the two groups (*p* 0.338). The two groups were compared in terms of PFS at the time of follow-up according to the baseline characteristics. Table 2 summarizes the results.

A significantly higher PFS for patients belonging to group B was observed in histological grades 1–2 (the median PFS was 92 months for group A vs. not reached for group B, *p* 0.046), as well as in 3a (the median PFS was 92 months vs. 122 for groups A and B, respectively, *p* 0.031). Regarding Ann Arbor Stage, a favorable trend was observed in stages I–II: the median PFS was 92 months in Group A vs. 122 months in Group B (*p* 0.074). On the other hand, comparing stages III–IV, a significant difference was shown in favor of the patients in group B with an estimated median PFS of 70.4 months in Group A vs. not reached in Group B (*p* 0.014). Among patients without Bulky disease, a significant difference in terms of PFS was observed between the two groups (the median PFS was 56.1 months for Group A vs. 122 months in Group B, *p* < 0.001); contrariwise, in the presence of bulky disease, a comparable outcome was observed (*p* 0.456), even if only 12 patients were assessed (Figure 2).

A favorable long-term PFS in group B was also observed in patients without BM involvement at diagnosis (PFS of 92 months vs. 121.8 months for group A and B respectively, *p* 0.003). This advantage was not confirmed for patients with involved BM (*p* 0.855) (Figure 2). A significantly better PFS was observed in Group B in the absence of B symptoms (median PFS 70 months vs. 121 months, *p* 0.001), extra-nodal involvement (median PFS 92 months vs. 121.8 months; *p* 0.001), augmented β2-microglobulin, and augmented LDH levels as shown in Table 2.

The Ki67 percentage regarding the immunohistochemistry was collected in 86% of our cohort. The median Ki67% in group A was 20 (range 10–70) and, in group B, was 30 (range 5–75) (*p* 0.021).

A correlation between the PFS and baseline SUV_max_ was observed in patients in different FLIPI risk categories. Forty-three (48.3%) patients were classified as low risk (median PFS 121 months), and 58.1% of them presented SUV_max_ > 6. The observed report is statistically significant, the median PFS was 92 months in Group A and 121.8 months in group B (*p* 0.012). No significant association between the PFS and baseline SUV_max_ was observed in the subgroup of intermediate (*p* 0.180) and high-risk FLIPI (*p* 0.347). A lower median PFS (99.7 months) was observed in high-risk patients compared to low risk.

Regarding the FLIPI2 score, most of the patients considered were low-risk (66.7%; median PFS 121.8 months). The difference in the median PFS in Group A and B was statistically significant (median PFS 92 months vs. 121.8 months, *p* 0.021). As the risk category increased, the median PFS was reduced; however, within each subgroup, patients with SUV_max_ <6 showed a lower median PFS (intermediate risk: estimated median PFS 70.4 months vs. not reached; high risk: 33.5 months vs. 38.3 months). The difference was not significant in patients with intermediate/high risk (*p* 0.137 and *p* 0.327 respectively). The results are shown in Figure 3.

We then compared the PFS in univariate analysis in the two groups based on the different induction therapies performed, and the results are shown in Table 3.

All patients received first-line treatment, eight of them after a W&W strategy lasting a median of 30 months (range 6–80 months). Three were in group A, and five were in group B; the SUV_max_ at baseline PET was considered for these patients. Fifteen patients in group A (44%) and 37 patients in group B (61%) received Rituximab maintenance.

Regarding I line therapy, 22 patients (23%) received radiotherapy exclusively and showed a 92-month global median PFS. The estimated median PFS was 92 months and was not reached in the two groups, respectively (*p* 0.124). In patients treated with R-CHOP (28.7%), the median PFS was 121.8 months. Nine patients were in group A (33.3%), and 18 were in group B (66.7%). The PFS was not statistically different in the two groups (*p* 0.278).

Thirty-four patients (36.1%) were treated with Rituximab and Bendamustine, and 27 (79.4%) were in group B. Although the results were not statistically significant, our analysis showed a trend with a better PFS in patients with a higher SUV_max_ at basal PET (*p* 0.062). Based on the end of treatment PET/CT, 78 patients (90.8%) achieved a CR: 26 with SUV_max_ ≤ 6 (33.3%) and 52 with SUV_max_ 6 (66.7%). Only eight patients achieved a PR (9.2%): four in group A and four in group B. 

A possible interference of Rituximab maintenance on the prognostic role of basal SUV_max_ was excluded (PFS within group A and group B was not different according to maintenance administration: *p* = 0.43 for group A and *p* = 0.22 for group B). In patients who achieved CR after the first-line treatment, the median PFS was significantly lower in the group with SUV_max_ ≤ 6 compared with in the group with SUV_max_ > 6 (PFS 92 months vs. 121.8 months; *p* 0.007) (Figure 1). In the PR subgroup, two relapses were observed in group A and one in group B: due to the exiguity of sample size, no significant difference was observed (*p* 0.364).

POD24 was not statistically different in the two groups, (Group A 3.2% vs. Group B 2.7%). We observed one event before 24 months in group A (1/34) and two events in the group B (2/60). Only 3 out of 94 patients (3.2%) underwent an aggressive transformation during the observation period: two in group A and one in group B. The median PFS was 40 months.

Finally, the multivariate analysis carried out by simultaneously analyzing all the parameters described above: the presence of a basal SUV_max_ > 6 as an independent favorable prognostic factor for PFS (OR 0.234; 95% IC 0.58–0.934; *p* 0.04) and the correlation between BM involvement at diagnosis and unfavorable prognosis (OR 5.98; 95% IC 1.5–23.3; *p* 0.011).

Considering these results, we further explored the outcome in terms of the PFS of group A and B based on the presence of at least two specific baseline characteristics considered as an indicator of more aggressive disease (bone marrow involvement, elevated LDH, elevated β2-microglobulin, extra-nodal disease, bulky disease, and the presence of B-symptoms). Therefore, patients were further classified as (A) patients with SUV_max_ <6 and no risk factors (26 Pts); (B) patients with SUV_max_ <6 and at least two risk factors (8 Pts); (C) patients with SUV_max_ > 6 and no risk factors (42 Pts); and (D) patients with SUV_max_ > 6 and at least two risk factors (18 Pts). The results are shown in Figure 4.

## 4. Discussion

The role of basal PET/CT as a predictor of PFS and OS in patients with FL has recently gained attention. Several studies have shown that elevated TMTV and total lesion glycolysis (TLG), obtained from the staging PET-TC, have a negative prognostic role, especially in patients with a high-tumor-burden disease [11,14,25].

The TMTV computation considers the SUV_max_ data together with the SUV threshold and disease extension and offers a promising advance on existing surrogates for tumor burden but potentially overestimates the volume of lesions with low SUV_max_, particularly for smaller volumes of interest [14,25].

In our analysis, SUV_max_ was chosen for its relatively simple assessment and its better reproducibility in comparison with more complex morpho-metabolic parameters, such as TMTV, which is less used in clinical practice. Despite expecting that a high SUV_max_ could identify patients at high risk, the results showed a different relationship: our observation demonstrates that the maximal FDG uptake considered as a single parameter retained an opposite prognostic significance to TMTV in a subset of patients. Although, in our analysis, no significant differences were found between the two subgroups in terms of OS, remarkably, a better long-term PFS was demonstrated in patients with a SUV_max_ greater than 6. We chose a cut-off with a better ratio between sensitivity and specificity on a ROC. A validation cohort is indeed needed to validate the chosen cut-off.

This does not conflict with the role of TMTV highlighted in the literature, as SUV_max_ does not take into account the extent and the size of the disease. Moreover, the SUV_max_ did not differ significantly between patients with higher or lower TMTV [11]. Previously published studies analyzed the prognostic role of SUV_max_ in terms of the risk of transformation and POD24. Although SUV cut-offs of 10, 14, 17, and 18 have been proposed to reveal transformation, in most of the studies proposed, a significant share of transformation in DLBCL was associated with SUVs below the chosen cut-off [13,26].

Univariate and multivariate analysis showed that the better prognostic impact of SUV_max_ was evident in the absence of other well-known risk factors, such as bone marrow involvement, Bulky disease, extra-nodal disease, elevated serum levels of Beta2, and LDH, as shown in Table 2. This result was confirmed by the univariate analysis conducted on the FLIPI and FLIPI2 scores, which proves that the advantage was mostly relevant in low-risk patients (Figure 3).

High metabolic activity at the onset, in the absence of other aggressive disease characteristics, was significantly correlated with a higher chance of long-term remission after treatment. Figure 4 shows that patients with SUV_max_ > 6 and carrying at least two risk factors had a comparable outcome to patients with SUV_max_ ≤ 6. Unexpectedly, patients who showed no risk factors at diagnosis and a baseline SUV_max_ ≤ 6 had a greater tendency to long-term relapse showing a worse outcome compared with patients who presented at least two aggressive disease characteristics and a high FDG uptake.

This finding is not related to a higher histological grading since no difference in terms of PFS was observed comparing grades 1–2 vs. 3a. The outcome in the subgroups was independent of the centroblast percentage, as well as from the Ann Arbor stage or number of nodal sites. Unfortunately, our sample size was not sufficient to determine a possible influence of the chemotherapy regimen employed (Table 3). The great majority of our patients achieved a CR after induction. Once more, patients with a higher FDG uptake who achieved a CR showed an increased long-term PFS (Figure 1). Such an outcome was not reproduced in patients who achieved a PR, implying that the persistence of disease at EOT-PET/CT, in patients with a basal increased SUV_max_, still retained an unfavorable prognostic impact.

Interestingly, the PFS in the two groups appeared to differ significantly only after 2 years of follow-up. The POD24 rate was, indeed, similar in the two groups. In recent studies that showed a higher rate of POD24 in patients with a high SUV_max_, most of the studied population was in an advanced stage at diagnosis and was classified as intermediate or high risk according to FLIPI score [27]. In our analysis, 40.4% of patients were in a limited stage, and 43% were classified as low-risk FLIPI (Table 1) and were equally distributed in the two groups. A possible hypothesis is that an increased FDG-uptake, reflecting also a higher Ki67 could be related to a higher proliferation rate, and consequently, a higher chemosensitivity (a figure more similar to aggressive NHL). 

Considering the correlation between basal SUV_max_ and tumor cell proliferating activity (Ki67), we can hypothesize a predominant role of tumor cells while the participation of the micro environment in the FDG-uptake should be further investigated [28]. Conversely, a more indolent disease can preserve a clone with a survival advantage and a prolonged relapse probability over time. In this specific case, the use of methods for detecting minimal residual disease (MRD) could be critical for identifying those patients initially classified as low risk but with a high propensity to relapse [25,26,29]. Finally, our study has several limitations, such as the retrospective nature, the limited sample size, and that SUV_max_ is susceptible to noise artifacts as it relies on a single voxel representation. Possibly, the joint use of other metabolic parameters, such as TMTV or delta-SUV, could be implemented in future studies to avoid these limitations in larger prospective studies.

## 5. Conclusions

In conclusion, our data demonstrated the independent prognostic role of baseline SUV_max_ as a PFS predictor, especially in patients with low-risk follicular lymphoma. Even without other risk factors, patients with low tumor metabolic activity exhibited a higher long-term relapse probability. Baseline SUV_max_ evaluation, with its simple assessment, could help identify patients at risk for late relapse, requiring strict follow-up and, potentially, MRD monitoring.

## Figures and Tables

**Figure 1 cancers-13-02876-f001:**
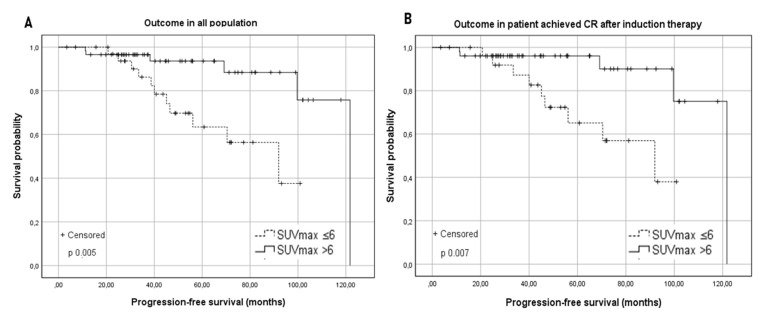
(**A**) Kaplan–Meier survival curves for the progression-free survival (PFS) in the studied population according to the baseline SUV_max_ with a cut-off of 6. (**B**) Kaplan–Meier survival curves for the progression-free survival (PFS) according to the baseline SUV_max_ in patients who achieved a complete response (CR) after induction therapy.

**Figure 2 cancers-13-02876-f002:**
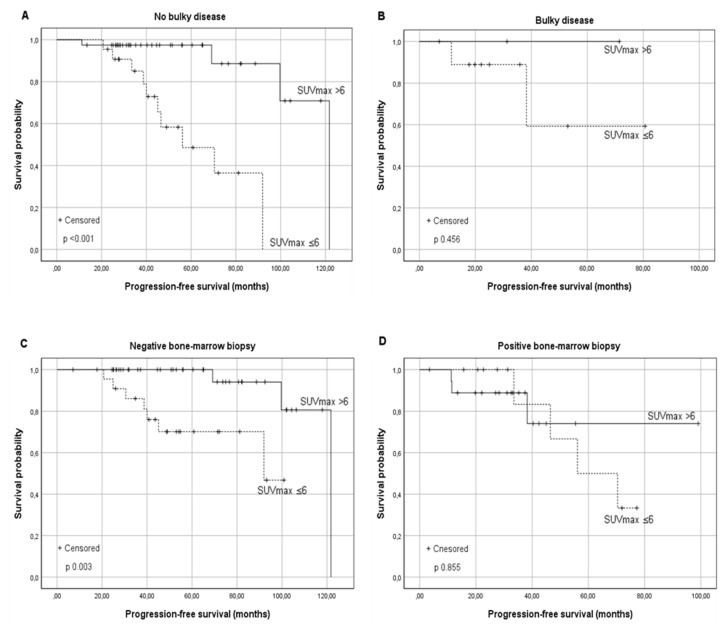
Kaplan–Meier survival curves for the progression-free survival (PFS) according to the baseline SUV_max_ in relationship to risk factors: (**A**) FL without Bulky disease, (**B**) FL with Bulky disease, (**C**) FL without Bone Marrow involvement, and (**D**) FL with Bone Marrow involvement. Similar results were found in the presence and absence of other risk factors reported in Table 2.

**Figure 3 cancers-13-02876-f003:**
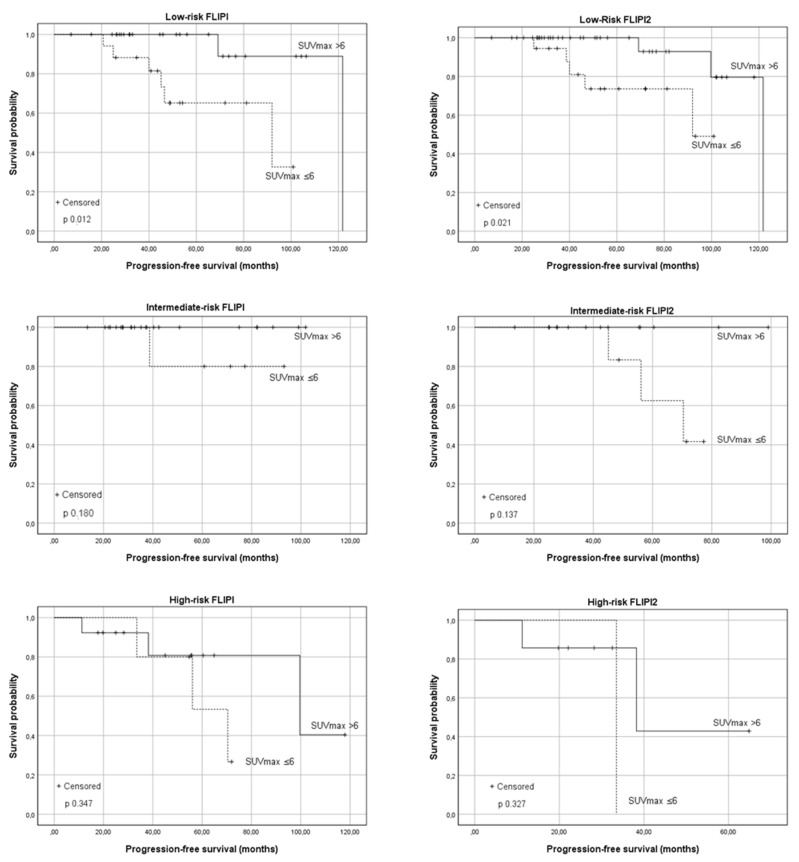
The progression-free survival (PFS) according to the baseline SUV_max_ in relationship to the Follicular Lymphoma International Prognostic Index (FLIPI) and Follicular Lymphoma International Prognostic Index 2 (FLIPI2) risk categories. From top to bottom: low-risk patients, intermediate-risk patients, and high-risk patients.

**Figure 4 cancers-13-02876-f004:**
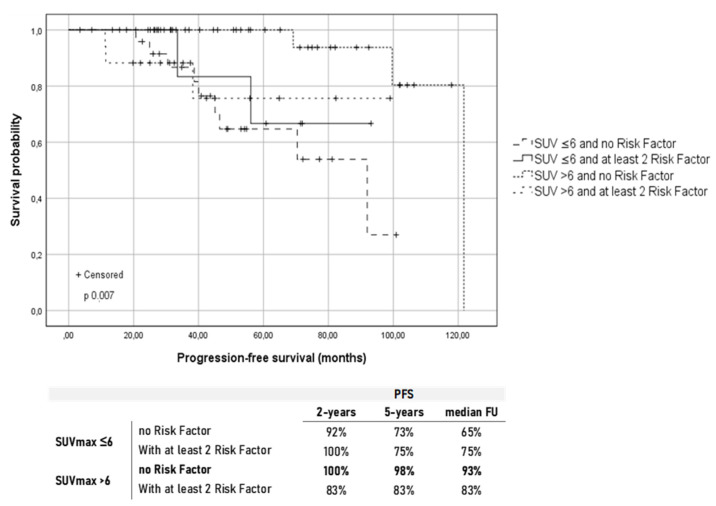
Kaplan–Meier survival curves for the progression-free survival (PFS) according to the baseline SUV_max_ and presence of at least two risk factors between bone marrow involvement, elevated LDH, elevated b2-microglobulin, extra-nodal disease, bulky disease, and presence of B-symptoms. Under the graphic are reported the PFS rates at different time points in the four groups.

**Table 1 cancers-13-02876-t001:** The baseline patient characteristics for the studied population, stratified according to the pretreatment Maximum Standardized Uptake Value (SUV_max_) with the cut-off at 6.

	All Patients	Group A	Group B	*p* Value *
	Tot = 94 N° (%)	Tot = 34 N° (%)	Tot = 60 N° (%)	
Histological Grading				0.409
1–2	47 (50)	21 (44.7)	26 (55.3)	
3a	41 (43.6)	11 (26.8)	30 (73.2)	
Undetermined	5 (5.3)	2 (40)	3 (60)	
Ann Arbor Stage				0.01
I–II	38 (40.4)	19 (50)	19 (50)	
III–IV	56 (59.6)	15 (26.8)	41 (73.2)	
Bulky Disease				0.191
No	61 (83.6)	22 (36.1)	39 (63.9)	
Yes	12 (16.4)	2 (16.7)	10 (83.3)	
Bone Marrow Involvement				0.869
Negative	63 (67.7)	22 (34.9)	41 (65.1)	
Positive	30 (32.3)	11 (36.7)	19 (63.3)	
Extranodal Disease				0.204
No	56 (60.2)	17 (30.4)	39 (69.6)	
Yes	37 (39.8)	16 (43.2)	21 (56.8)	
B Symptoms				0.496
No	82 (88.2)	31 (37.8)	51 (62.2)	
Yes	11 (11.8)	3 (27.8)	8 (72.2)	
N° Nodal Sites				0.065
≤3	34 (38.6)	16 (47.1)	18 (52.9)	
>3	54 (61.4)	15 (27.8)	39 (72.2)	
β2-Microglobulin				0.252
≤ULN	64 (82.1)	24 (37.5)	42(62.5)	
>ULN	14 (17.9)	3 (21.4)	11 (78.6)	
LDH				0.367
≤ULN	70 (85.4)	27 (38.6)	43 (61.4)	
>ULN	12 (14.6)	3 (25)	9 (75)	
FLIPI				0.57
Low risk	43 (48.3)	18 (41.8)	25 (58.1)	
Intermediate risk	28 (31.5)	10 (35.7)	18 (64.3)	
High risk	18 (20.2)	5 (27.8)	13(72.2)	
FLIPI 2				0.36
Low risk	56 (66.7)	20 (35.7)	36 (64.3)	
Intermediate risk	20 (23.8)	8 (40)	12 (60)	
High risk	8 (9.5)	1 (12.5)	7 (87.5)	

Group A: SUV_max_ ≤ 6; Group B: SUV_max_ > 6; β2-M: β2-Microglobulin; LDH: lactate dehydrogenase; FLIPI: Follicular Lymphoma International Prognostic Index; and FLIPI2: Follicular Lymphoma International Prognostic Index 2. * Categorical covariates were compared to the two groups using Fisher’s exact test or the Chi^2^ test.

**Table 2 cancers-13-02876-t002:** Comparison of the PFS between Group A and Group B according to each baseline characteristic.

	PFS at Time of Follow-Up			*p* Value *
	All Patients N° (%)	Group A N° (%)	Group B N° (%)	
Histological Grading				
1–2	37 (78.7)	14 (66.7)	23 (88.5)	0.046
3a	34 (82.9)	7 (63.6)	27 (90)	0.031
Undetermined	5 (100)	2 (100)	3 (100)	
Ann Arbor Stage				
I–II	31 (81.6)	14 (73.7)	17 (89.5)	0.075
III–IV	46 (82.1)	9 (60)	37 (90.2)	0.014
Bulky Disease				
No	47 (77)	12 (54.5)	35 (89.7)	<0.001
Yes	10 (83.3)	2 (100)	8 (80)	0.456
Bone Marrow Involvement				
No	53 (84.1)	15 (68.2)	38 (92.7)	0.003
Yes	23 (76.7)	7 (63.6)	16 (84.2)	0.855
Extranodal Disease				
No	47 (83.9)	11 (64.7)	36 (92.3)	0.001
Yes	29 (78.4)	11 (68.8)	18 (85.7)	0.855
B Symptoms				
No	66 (80.5)	20 (64.5)	46 (90.2)	0.001
Yes	10 (90.9)	3 (100)	7 (87.5)	0.386
N° Nodal Sites				
≤3	29 (85.3)	12 (75)	17 (94.4)	0.028
>3	43 (79.6)	9 (60)	34 (87.2)	0.059
Β2-Microglobulin				
≤ULN	52 (81.3)	16 (66.7)	36 (90)	0.015
>ULN	12 (85.7)	2 (66.7)	10 (90.9)	0.307
LDH				
≤ULN	58 (82.9)	19 (70.4)	39 (90.7)	0.026
>ULN	9 (75)	1 (33.3)	8 (88.9)	0.093
FLIPI				
Low risk	35 (81.4)	12 (66.7)	23 (92)	0.012
Intermediate risk	27 (96.4)	9 (90)	18 (100)	0.180
High risk	12 (66.7)	2 (40)	10 (76.9)	0.347
FLIPI 2				
Low risk	48 (62.5)	15 (75)	33 (91.7)	0.021
Intermediate risk	17 (85)	5 (62.5)	12 (100)	0.137
High risk	5 (62.5)	0 (0)	5 (71.4)	0.327

PFS: Progression-free survival; Group A: SUV_max_ ≤ 6; Group B: SUV_max_ > 6; β2-M: β2-Microglobulin; LDH: lactate dehydrogenase; FLIPI: Follicular Lymphoma International Prognostic Index; and FLIPI2: Follicular Lymphoma International Prognostic Index 2. * The Kaplan–Meier method was used to compare the two groups in terms of PFS, significant findings are typed bold.

**Table 3 cancers-13-02876-t003:** Comparison of the PFS between Group A and Group B in relationship to I line therapy adopted and response achieved at the end of induction. Sub-analysis was performed for those patients with transformation to high-grade lymphoma at relapse.

Treatment/Response	All Patients	Group A	Group B	*p* Value *
	DF-Pts/Tot (%)	DF-Pts/Tot (%)	DF-Pts/Tot (%)	
I line Therapy				
Radiotherapy	16/22 (72.7)	7/12 (58.3)	9/10 (90)	0.124
R-CHOP	22/27 (81.5)	7/9 (77.8)	15/18 (83.3)	0.278
R-Benda	30/34 (88.2)	5/7 (71.4)	25/27 (92.6)	0.062
Other therapies	7/11 (66)	4/7 (57)	3/4 (75)	0.480
Response after induction				
CR	65/78 (83.3)	17/26 (65.4)	48/52 (90.3)	0.007
PR	5/8 (62.5)	2/4 (50)	3/4 (75)	0.364
Transformation at relapse				
No	77/91 (84.6)	23/32 (71.9)	54/59 (91.5)	0.008
Yes	0/3	0/2	0/1	0.225

PFS: Progression-free survival; Group A: SUV_max_ ≤ 6; Group B: SUV_max_ > 6; DF-pts: Disease-free patients at time of follow-up; R-CHOP: Rituximab, Cyclophosphamide, Vincristine, Daunorubicin, Prednisone; R-Benda: Rituximab and Bendamustine; CR: Complete remission; and PR: Partial remission. * The Kaplan–Meier method was used to compare the two groups in terms of PFS; significant findings are in bold.

## Data Availability

No new data were created or analyzed in this study. Data sharing is not applicable to this article.

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
