# Peer review of "High Basal Maximal Standardized Uptake Value (SUVmax) in Follicular Lymphoma Identifies Patients with a Low Risk of Long-Term Relapse"

_cancers, 2021, doi:10.3390/cancers13122876_

Round 1

Reviewer 1 Report

I would like to congratulate the authors for the manuscript. It is an elegant and practical idea to employ a widely available data to predict relapse in follicular lymphoma and identify patient at higher risk for relapse. The authors retrospectively analyse a cohort of follicular lymphoma cases evaluating initial SUVmax as a prognostic factor for progression free survival. They found that the value of SUVmax greater than 6 is associated with a lower risk of relapse independently of other risk factors as bone marrow involvement, present of B-symptoms. The retrospective study setting and the subject number are appropriate to evaluate the questions. The preparation of the manuscript is properly performed. (Few typos could be seen as in line 250 and 269)

I felt that the following minor points should be addressed before the manuscript is accepted:

  1. Does the presence of maintenance treatment interfere with the outcome? Post-induction maintenance treatment was not evaluated as a factor and there is no detail in the Methods section on the maintenance treatment.
  2. It is not clear why the cut of value 6 was chosen to create the two subgroups. Is there a statistical evidence to support the choice?
  3. The group of patients that was treated with a WnW strategy is less clearly described. Was the SUVmax value collected at the time of diagnosis or at the time of treatment initiation? How were these patients evaluated according to the SUVmax value?

I highly recommend the manuscript for acceptance for publication after these minor queries are addressed.

Author Response

Reviewer 1

Comments and Suggestions for Authors

I would like to congratulate the authors for the manuscript. It is an elegant and practical idea to employ a widely available data to predict relapse in follicular lymphoma and identify patient at higher risk for relapse. The authors retrospectively analyse a cohort of follicular lymphoma cases evaluating initial SUVmax as a prognostic factor for progression free survival. They found that the value of SUVmax greater than 6 is associated with a lower risk of relapse independently of other risk factors as bone marrow involvement, present of B-symptoms. The retrospective study setting and the subject number are appropriate to evaluate the questions. The preparation of the manuscript is properly performed. (Few typos could be seen as in line 250 and 269)

I felt that the following minor points should be addressed before the manuscript is accepted:

Dear Reviewer 1,

Thank you for your suggestions. Here we provide a point-by-point response to your comments:

  1.  Does the presence of maintenance treatment interfere with the outcome? Post-induction maintenance treatment was not evaluated as a factor and there is no detail in the Methods section on the maintenance treatment.

Answer: We verified a possible effect due to maintenance with rituximab: patients who underwent maintenance were mostly patients treated with rituximab-chemotherapy and were 44% in group A and 61% of group B. (Maintenance was not administered in patients with early stage FL and in a minority of advanced stage according to clinical judgement.) We performed an univariate analysis (Pearson Chi-square test) to evaluate a possible impact of maintenance according to SUVmax in each group, no significant difference was demonstrated, p=0.43 for group A and p=0.22 for group B. Concluding that rituximab maintenance does not interfere in the prognostic role of SUVmax. This point was clarified in the Methods (line 139-141) and Results (line 270-272, 282-285) sections.

  1. It is not clear why the cut of value 6 was chosen to create the two subgroups. Is there a statistical evidence to support the choice?

Answer: We firstly performed a ROC curve using the value of SUVmax and the event of relapse. The area of the ROC curve was 0.672 with a p 0.036; we chose a cut-off of 6 showing the best ratio between sensitivity and specificity (60% and 73% respectively) and the highest value of significance with Kaplan-Meier method (p 0.005) figure 1. Increasing the cut off above 7 would have considerably decreased specificity. This point was clarified in the methods section (line 123-130).

  1. The group of patients that was treated with a WnW strategy is less clearly described. Was the SUVmax value collected at the time of diagnosis or at the time of treatment initiation? How were these patients evaluated according to the SUVmax value?

Answer: Three patients with basal SUVmax ≤6 and 5 patients with SUVmax >6 were allocated to a watch and wait policy. The duration of W&W phase was 30 months ranging from 6 to 80 months with no difference between the two groups. The data numbers are too small but basal SUV seems not influence the duration of the time of treatment initiation. This point was clarified in the Results section (Line 268-271)

Reviewer 2 Report

Assanto at al conducted an analysis of SUVmax parameter in FL at diagnosis to evaluate its prognostic impact and the risk of late relapse, which is an interesting question. They conclude that higher basal FDG uptake (>6) were associated with a higher PFS in absence of FLIPI risks factors.

The manuscript is well-written and presented. However, this study is exclusively descriptive and the results are counter-intuitive and, above all, they are not consistent with other recent publications (Rossi C and al, Haematologica 2020 ; Strati P, Haematologica 2020), that are poorly or little discussed.

In general, I also think that the manuscript would benefit from a more deep discussion of the results and validation on another cohort of patients.   

Major comments:

- At first, does Suvmax> 6 mean “high” Suvmax or strong Suvmax induction? it's not clear in the text and confusing in the figure.

- It is not described how groups A and B were developed (SUV max> 6 for group A and <6 for B? As the legend of Table 1 suggests). This needs to be clarified in the text.

- If Suvmax> 6 means “high” Suvmax, the threshold chosen is much lower than those already published, the comments on this point are insufficient and must be substantiated. If done, the ROC curves should be shown.

- there is no clear information about POD24, this should be added in the results and not just at the end of the discussion, and should be discuss further.

- There is no exploration of biological data (except those included in FLIPI score), do you have immunohistochemistry data for example (BCL2, KI67, immune infiltration…)?

Morevover, influence of mutation profiles (NGS or at least major mutations (KMT2D, CREBBP, BCL2, FOXO1)) must be determined due to their implication in progression but also in baseline SUVmax (Rossi C, Haematologica 2020). This latter publication should also be cited and discussed.

- The authors discard comments on therapies because of the too small size of their cohort. As results may remain controversial, it must be confirmed on another cohort, and evaluate against first line treatment and especially rituximan treatment.  

- It is unfortunate that authors have no evaluation of TMTV, which is a strong predictor of outcome. This should be added.

Minor comments:

- Line 41: FLIPI 21 probably means FLIPI 2 with reference 1. By the way, references in all the text are not placed in bracket or superscript. Please modify to improve readability.

- Does patient with watch & way strategy be equally divided in group A and B?

- Table 3: below the table, indications seem to belong to the legend of the table, it must be written smaller so not to interfere with the reading of the main text.

- Hypothesis to explain “high” Suvmax at baseline should be better discussed (mediated by tumor cells ? and/or microenvironment ? )

- It would be interesting to discuss the interest of the Δ SUVMax determination in this context.

Author Response

Rev 2

Comments and Suggestions for Authors

Assanto at al conducted an analysis of SUVmax parameter in FL at diagnosis to evaluate its prognostic impact and the risk of late relapse, which is an interesting question. They conclude that higher basal FDG uptake (>6) were associated with a higher PFS in absence of FLIPI risks factors.

The manuscript is well-written and presented. However, this study is exclusively descriptive and the results are counter-intuitive and, above all, they are not consistent with other recent publications (Rossi C and al, Haematologica 2020; Strati P, Haematologica 2020), that are poorly or little discussed.

In general, I also think that the manuscript would benefit from a more deep discussion of the results and validation on another cohort of patients.

Dear Reviewer, 2,

Thank you for your suggestions. Here we provide a point-by-point response to your comments:

Major comments:

- At first, does Suvmax> 6 mean “high” Suvmax or strong Suvmax induction? it's not clear in the text and confusing in the figure.

- If Suvmax> 6 means “high” Suvmax, the threshold chosen is much lower than those already published, the comments on this point are insufficient and must be substantiated. If done, the ROC curves should be shown.

- It is not described how groups A and B were developed (SUV max> 6 for group A and <6 for B? As the legend of Table 1 suggests). This needs to be clarified in the text.

Answer: Patients were stratified according to the Standardized FDG Uptake Value at baseline, in the highest metabolically active lesion. To identify the best SUVmax cutoff for our analysis, we firstly performed a ROC curve using the value of SUVmax and the event of relapse. The area of the ROC curve was 0.672 with a p 0.036; we chose a cut-off of 6 showing the best ratio between sensitivity and specificity (60% and 73% respectively) and the highest value of significance with Kaplan-Meier method (p 0.005) figure 1. Increasing the cut off above 7 would have considerably decreased specificity. This point was clarified in the Methods section (line 123-130). Two groups of patients were defined: Group A with a lower SUVmax (≤ 6) and Group B characterized by a SUVmax > 6. This point was clarified in the text and in the legend of the figure (line 123-130, legend of table 1 and 3).

Already published studies analysed the prognostic role of SUVmax in terms of accuracy in identifying risk of transformation (Schöder H, J Clin Oncol. 2005; Karam M Nucl Med Commun. 2011) and POD24 (Strati et al, Haematologica 2020). Although SUV cut-offs of 10, 14, 17 and 18 have been proposed to reveal transformation, in most of the studies proposed, a significant share of transformation in DLBCL was associated with SUVs below the chosen cut-off. Nevertheless, the purpose of our study was not to analyse the risk of transformation (which was extremely low in our case series: 2 patients in Group A and 1 patient in Group B) but to identify a possible prognostic role of different levels of metabolic activity among FL patients. This point is clarified in the Discussion section (Line 350-354)

- there is no clear information about POD24, this should be added in the results and not just at the end of the discussion and should be discuss further.

Answer: POD24 was not statistically different in the two groups: We observed 1/34 (3.2%) event before 24 months in group A and 2/60 (2.7%) in group B. Of course, no significant difference could be observed with such a small number of events. These informations were included in the Results (Line 249-251) and in the Discussion (347-348). It should be mentioned that in recent studies showing a higher rate of POD24 in patients with high SUVmax, most of the studied population was in advanced stage at diagnosis and was classified as intermediate or high risk according to FLIPI score (Rossi C, Haematologica 2020; Strati Haematologica 2020). In our analysis 40.4% of patients were in limited stage and 43% classified as Low-risk FLIPI (table 1) and were equally distributed in the two groups. This point was added in the Discussion section (Line 386-395)

- There is no exploration of biological data (except those included in FLIPI score), do you have immunohistochemistry data for example (BCL2, KI67, immune infiltration…)?

Morevover, influence of mutation profiles (NGS or at least major mutations (KMT2D, CREBBP, BCL2, FOXO1)) must be determined due to their implication in progression but also in baseline SUVmax (Rossi C, Haematologica 2020). This latter publication should also be cited and discussed.

Answer: Patients were retrospectively analysed since 2008. Next Generation Sequencing and Gene Expression Profile were not available at that time, nor they are routinely performed even now in clinical practice given the high costs and their not yet clear role in terms of prognosis and choice of treatment. Although very interesting for a possible correlation with tumour metabolic activity and prognosis we have too limited data to draw any conclusion.

Regarding immunohistochemistry data, Ki67 percentage was collected in 86% of our population. Median Ki67% in group A was 20 (range 10-70), while in group B was 30 (range 5-75) (p 0.021). This finding support the assumption that patients with a higher FDG-uptake also exhibit a higher proliferation rate (Ki67%).  These data have been included in the Results section (line228-229), and Discussion section (line 390-395)

Rossi C et al showed that SUVmax was significantly increased in patients with higher percentage of Ki67 probably indicating a relationship between FDG uptake and cell proliferation. No significant relationship was instead found between SUVmax and microenvironment and T-cell infiltration. Our analysis confirmed an association between high Ki67 values ​​and higher SUVmax values. We can, therefore, speculate that the metabolic activity (and the consequent FDG avidity), in FL, is mainly related to tumor cell proliferation. Again, a validation cohort and a systematic review of the histological samples would better analyse this phenomenon.Discussion Section (Line 387-395)

- The authors discard comments on therapies because of the too small size of their cohort. As results may remain controversial, it must be confirmed on another cohort, and evaluate against first line treatment and especially rituximan treatment.  

Answer: Results at follow-up are shown in table III. In the R-Benda group, 2 out of 7 patients relapsed in group A and 2 out of 27 in group B, while in the R-CHOP events are 2/9 for group A and 3/18 for group B. As well for radiotherapy alone, 5 out of 12 patients relapsed in group A and 1 out 10 in group B, other treatments were Rituximab plus minor chemotherapy regimens or rituximab monotherapy after Radiotherapy. A more extensive study with a proper sample size, should be necessary to analyze the role of different treatments. We left unmodified the Discussion on this point (line 380-381)

- It is unfortunate that authors have no evaluation of TMTV, which is a strong predictor of outcome. This should be added.

Answer: Since TMTV evaluation was not the purpose of the study, a specific analysis in this direction was not performed. We agree that it would be exciting to investigate the prognostic value of SUVmax and TMTV. Actually, we are experiencing some issues due to the threshold of activity for TMTV computation in tumors with low FDG uptake, with a great TMTV being falsely detected in tumors with a poor activity. For these reasons TMTV is actually not used in our clinical routine. Furthermore, TMTV derives from a complex algorithm not fully standardized and requires the computation of the volumes of the disease. As elucidated in the Discussion section (line 398-401), the meaning of the isolate SUVmax information is quite different from TMTV with sometimes conflicting results but has the advantage to be easily available. This type of comparison could be the objective of a future study.

Minor comments:

- Line 41: FLIPI 21 probably means FLIPI 2 with reference 1. By the way, references in all the text are not placed in bracket or superscript. Please modify to improve readability.

Answer: Thank you we revised the paper to improve readability.

- Does patient with watch & way strategy be equally divided in group A and B?

Answer: Three patients with basal SUVmax ≤6 and 5 patients with SUVmax >6 were subsequently allocated to a watch and wait policy. The duration of W&W phase was 30 months ranging from 6 to 80 months with no difference between the two groups. The data numbers are too small but basal SUV seems not influence the duration of the time of treatment initiation. This point was clarified in the Results section (Line 268-271)

- Table 3: below the table, indications seem to belong to the legend of the table, it must be written smaller so not to interfere with the reading of the main text.

Answer: Thank you for your advice. We modified the tables and the legends accordingly.

- Hypothesis to explain “high” Suvmax at baseline should be better discussed (mediated by tumor cells? and/or microenvironment?)

Answer: This point was furtherly clarified in the text in the Discussion section (lines 387-397)

- It would be interesting to discuss the interest of the Δ SUVMax determination in this context.

Answer: The purpose of our study was to evaluate the prognostic role of SUVmax at baseline in terms of PFS and OS and not as an indicator of response to treatment. Evaluation of Δ SUVmax between baseline and EOT-PET would an interesting point concerning the quality of treatment response as well as many other possible parameters. This could be, definitely, a possible topic of a future research.

Reviewer 3 Report

The authors have evaluated the semi-quantative parameter SUVmax among 94 patients with follicular lymphoma. Overall this could be informative, but I have major concerns regarding the measurement of SUVmax, the determination of the cutt-off, inconsistencies in results and data presentation ,the lack of a plausible rationale.

Introduction:                                 
Identification of patients who have an unfavorable outcome in FL is relevant, but should improve on the FLIPI, FLIPI-2, m7-FLIPI or dynamic models. The authors introduce SUVmax as an potential easy read-out for prognostication. As described by the authors others have shown high SUVmax to be related to inverse outcome, but this is not hypothesized for the current study?

Material and methods:                               
Was informed consent given by the patients of was this waved by the METc?
Measurement of SUVmax was subject to 1 physician? This should have been done by at least two independent reviewers and checked for inter-observer variability.

How was the cut-off of 6 points determined? If based on ROC, then there should be a training an validation cohort.

I cannot quite follow the results. The numbers seem to be off?

Whole cohort (n=94) 5-year PFS 87.5% ~ 12 events

Group A (n=30) 5-year PFS 74.5 ~ 5 events

Groep B (n=60) 5-year PFS 95 ~ 3 events

Hence, I am missing 4 events?

When looking at figure 1 A in group A I only count 2 events until 5-year in the SUVmax < 6, while I count 9 events in the SUVmax > 6? The 5-year PFS in the latter group at 60 months is below 70%.

Subsequently a lot of variables are analyzed and combined with SUV results. It is fine to look at the prognostic value of individual variables (univariate analysis), but in order to state something about combination of results there needs to be a formal multivariate analyses. Furthermore, the authors should probably correct for multiple testing.

Discussion

There are limitatations to SUVmax, as it is susceptible to noise artefacts as it relies on single voxel representation. This should be addressed as a limitation.

Pathofysiologically it doesn't make sense that tumors with high metabolic rate (SUVmax) have the best long term survival? The suggestion that metabolic active tumors are more suspectable to chemotherapy is unproven and contradicts findings by other groups on SUVmax. If there is more data supporting this hypothesis, the authors should introduce this better.

Author Response

REV 3

Comments and Suggestions for Authors

Dear Reviewer 3,

Thank you for your suggestions. Here we provide a point-by-point response to your comments:

The authors have evaluated the semi-quantative parameter SUVmax among 94 patients with follicular lymphoma. Overall this could be informative, but I have major concerns regarding the measurement of SUVmax, the determination of the cutt-off, inconsistencies in results and data presentation ,the lack of a plausible rationale.

Introduction:                                 
Identification of patients who have an unfavorable outcome in FL is relevant, but should improve on the FLIPI, FLIPI-2, m7-FLIPI or dynamic models. The authors introduce SUVmax as a potential easy read-out for prognostication. As described by the authors others have shown high SUVmax to be related to inverse outcome, but this is not hypothesized for the current study?

Answer: The lack of rationale of our results and the apparently conflicting findings with respect to TMTV, is only apparent. Although SUVmax is one of the parameters considered in TMTV elaboration, when considered alone it deserves a different significance. While TMTV is mainly conditioned by the disease extension, SUVmax just reflects the punctual higher FDG-uptake. In the discussion section lines we analyse this point and try to produce an interpretation. (Lines 386-405)

We started the analysis hypothesizing an high metabolic activity could identify patients carrying an high risk disease. The results did not confirm this hypothesis, so we tried to better analyse the phenomenon concluding that a SUV max lower than 6 correlates with an inferior prognosis in terms of PFS mainly in long terms in low risk patients.

We then performed a ROC Curve using the value of SUVmax and the event of relapse. We firstly performed a ROC curve using the value of SUVmax and the event of relapse. we firstly performed a ROC curve using the value of SUVmax and the event of relapse. The area of the ROC curve was 0.672 with a p 0.036; we chose a cut-off of 6 showing the best ratio between sensitivity and specificity (60% and 73% respectively) and the highest value of significance with Kaplan-Meier method (p 0.005) figure 1. Increasing the cut off above 7 would have considerably decreased specificity. This point was clarified in the methods section (line 121-126). Increasing the cut off above 7 would have considerably decreased specificity. This point was clarified in the methods section (line 123-130). As shown in fig.3 and 4, PFS was significantly better in patients with SUVmax >6 with a PFS of 67.6% (11/34) for group A and 90% for group B (6/60) p 0.005 estimated median PFS 91 months vs 129.  POD24 was comparable in the two groups with a 2-Year-PFS 97% (1/34) for group A and 97% (2/60) for group B. This point was clarified in the  Methods section (line 123-130).

Material and methods:                               
Was informed consent given by the patients of was this waved by the METc?
Measurement of SUVmax was subject to 1 physician? This should have been done by at least two independent reviewers and checked for inter-observer variability.

Answer: An informed consent was signed by enrolled patients, and the form was produced as requested by the Ethical Committee, also a study approval by IRB was produced. We agree with the reviewer that SUVmax measurement could be subjected to inter observer variability. While SUVmax computation is relatively automatic, region of interest (ROI) placement should be carefully checked in order to exclude unwanted tissues in the area of interest.  This relevant aspect is now included in the materials and methods section. In particular, as already mentioned in other reports of our group in this field, after their positioning, all the VOIs were furthermore checked, in a three planar view, by two experienced physicians to exclude unwanted tissues in the area of interest. This point was clarified in the Methods section (line 113-116) and the limits reported in the Discussion (Line 400-405)

How was the cut-off of 6 points determined? If based on ROC, then there should be a training an validation cohort.

Answer: This point was already clarified above.

I cannot quite follow the results. The numbers seem to be off?

Whole cohort (n=94) 5-year PFS 87.5% ~ 12 events

Group A (n=30) 5-year PFS 74.5 ~ 5 events

Groep B (n=60) 5-year PFS 95 ~ 3 events

Hence, I am missing 4 events?

When looking at figure 1 A in group A I only count 2 events until 5-year in the SUVmax < 6, while I count 9 events in the SUVmax > 6? The 5-year PFS in the latter group at 60 months is below 70%.

Answer: We do apologise about an error in the numbers reported, that we corrected in the text. In fig 1A in Group A (SUVmax < = 6) 9 events out of 34 patients are reported. In group B (SUV max > 6) 3 events out of 60 patients. PFS at 60mo was 73.5% and 95% in groups A and B respectively. Line 189-193

Subsequently a lot of variables are analysed and combined with SUV results. It is fine to look at the prognostic value of individual variables (univariate analysis), but in order to state something about combination of results there needs to be a formal multivariate analyses. Furthermore, the authors should probably correct for multiple testing.

We performed a multivariate analysis using cox regression model: the reports what is written in the test line 309-313, the variables included were: Bone marrow biopsy, I line, SUVmax group, Bulky disease, stage, grading, FLIPI, FLIPI2, LDH altered and presence of B-symptoms

Discussion

There are limitatations to SUVmax, as it is susceptible to noise artefacts as it relies on single voxel representation. This should be addressed as a limitation.

Answer: We introduced this limitation of our study at the end of the discussion section (Line 400-405).

Pathofysiologically it doesn't make sense that tumors with high metabolic rate (SUVmax) have the best long term survival? The suggestion that metabolic active tumors are more suspectable to chemotherapy is unproven and contradicts findings by other groups on SUVmax. If there is more data supporting this hypothesis, the authors should introduce this better.

Answer: As already discussed, we were surprised by the obtained results. Our interpretation of the phenomenon is that a tumour with a lower FDG-uptake, as well as a lower Ki67 reflects an indolent behaviour with a higher tendency of long terms relapse. A more metabolically active tumour resembling an aggressive lymphoma behaviour, it is prone to relapse early or to be cured. Of course, this is a hypothesis of interpretation of the results obtained that we offer for discussion. Discussion Section (Line 387-395). Ki67 percentage was collected in 86% of our population. Median Ki67% in group A was 20 (range 10-70), while in group B was 30 (range 5-75) (p 0.021). This finding supports the assumption that patients with a higher FDG-uptake also exhibit a higher proliferation rate (Ki67%).  These data have been included in the Results section (line228-229), and Discussion section (line 390-395)

Rossi C et al showed that SUVmax was significantly increased in patients with higher percentage of Ki67 probably indicating a relationship between FDG uptake and cell proliferation. No significant relationship was instead found between SUVmax and microenvironment and T-cell infiltration. Our analysis confirmed an association between high Ki67 values and higher SUVmax values. We can, therefore, speculate that the metabolic activity (and the consequent FDG avidity), in FL, is mainly related to tumor cell proliferation. Again, a validation cohort and a systematic review of the histological samples would better analyse this phenomenon. Discussion Section (Line 387-395)

Round 2

Reviewer 2 Report

Thank you for your responses to my comments. They are cleary displayed.

In my opinion, your comments excluding TME participation to biological function is inappropirate and should be nuanced.

Further studies to confirm your results would be interesting...

Reviewer 3 Report

The authors addressed many of the raised questions. However, I do want to point out some items that need improvement.

Introduction
The authors introduce the negative predictive value by others of TMTV (lines 58-67) and SUVmax (lines 68-82) in FL. However, I would really like to have the authors describe their hypothesis as they did in the rebuttal ‘We started the analysis hypothesizing an high metabolic activity could identify patients carrying an high risk disease’. Now the last paragraph of the introduction is still not clear on that. Furthermore there is a lot of redundancy in this paragraph (The aim of the study was to study the prognostic value of SUVmax at baseline with respect to OS and PFS in patients with FL)

Material and methods:
- The ICF of patients is still missing. You can have a waiver by a METC or you can have an approval and still need an ICF of patients. Please add this information.

-Control of VOI is acceptable

- ROC cut-off. This remains problematic. The characteristics are actually pretty poor (60% sensivity and 73% specificity). And while you can calculate the optimal ROC cut-off for this serie, you really need a validation cohort to determine if your cut-off is valid, especially with the limited number of events in this series and the unexpected outcome (which is different from what other groups have reported). Given that it is unlikely the authors can do a validation, this limitation should clearly be addressed in the discussion (The results are therefore more hypothesis generating)

Results:
- data inconsistencies were solved

- Univariate analysis still contains a lot of unnecessary graphs (2A-D and figure 3). It would be much more clean and informative to provide a table for univariate analysis, rather than showing a lot of graphs with two variable (e.a. SUVmax plus bulk, bone marrow). Subsequently report on the multivariate analysis (based on the factors associated in univariate analysis) which is now tugged away. And if there are more variable significantly associated with survival you can combine these into a model and show the graph (figure 4)
- Question for correcting for multiple testing was not addressed.

Discussion:
- Anything without a plausible mechanism and validation cohort should be interpreted with caution. As stated by the authors themselves ‘As already discussed, we were surprised by the obtained results’ I think this should be adressed at the beginning of the discussion. The hypothesis was that high SUVmax was inversely related to outcome. Surprisingly they found an opposite relation.
